# Limited Response of Indigenous Microbes to Water and Nutrient Pulses in High-Elevation Atacama Soils: Implications for the Cold–Dry Limits of Life on Earth

**DOI:** 10.3390/microorganisms8071061

**Published:** 2020-07-16

**Authors:** Lara Vimercati, Clifton P. Bueno de Mesquita, Steven K. Schmidt

**Affiliations:** 1Department of Ecology and Evolutionary Biology, University of Colorado, Boulder, CO 80309-0334, USA; Lara.Vimercati@Colorado.edu (L.V.); cliff.buenodemesquita@colorado.edu (C.P.B.d.M.); 2Institute of Arctic and Alpine Research, University of Colorado, Boulder, CO 80309-0450, USA

**Keywords:** Atacama, extremophiles, *Naganishia friedmannii*, high elevation, astrobiology, resource limitation, Volcán Llullaillaco, extreme environment, microbial community, soil microcosm

## Abstract

Soils on the world’s highest volcanoes in the Atacama region represent some of the harshest ecosystems yet discovered on Earth. Life in these environments must cope with high UV flux, extreme diurnal freeze–thaw cycles, low atmospheric pressure and extremely low nutrient and water availability. Only a limited spectrum of bacterial and fungal lineages seems to have overcome the harshness of this environment and may have evolved the ability to function in situ. However, these communities may lay dormant for most of the time and spring to life only when enough water and nutrients become available during occasional snowfalls and aeolian depositions. We applied water and nutrients to high-elevation soils (5100 meters above sea level) from Volcán Llullaillaco, both in lab microcosms and in the field, to investigate how microbial communities respond when resource limitations are alleviated. The dominant taxon in these soils, the extremophilic yeast *Naganishia* sp., increased in relative sequence abundance and colony-forming unit counts after water + nutrient additions in microcosms, and marginally in the field after only 6 days. Among bacteria, only a *Noviherbaspirillum* sp. (Oxalobacteraceae) significantly increased in relative abundance both in the lab and field in response to water addition but not in response to water and nutrients together, indicating that it might be an oligotroph uniquely suited to this extreme environment. The community structure of both bacteria and eukaryotes changed significantly with water and water + nutrient additions in the microcosms and taxonomic richness declined with amendments to water and nutrients. These results indicate that only a fraction of the detected community is able to become active when water and nutrients limitations are alleviated in lab microcosms, and that water alone can dramatically change community structure. Our study sheds light on which extremophilic organisms are likely to respond when favorable conditions occur in extreme earthly environments and perhaps in extraterrestrial environments as well.

## 1. Introduction

Recent research indicates that some of the most extreme soil ecosystems on Earth are found at high elevations in the dry valleys and slopes of the high Andes [1,2,3,4]. In fact, the driest high mountains on Earth occur just to the east of the Atacama Desert, where numerous, massive stratovolcanoes rise from the Puna de Atacama [1]. Geographical barriers in this area restrict the flow of atmospheric moisture, which in turn results in some of the most inhospitable proto-mineral soils on the planet, which contain nearly undetectable levels of soil water, organic carbon stocks, microbial biomass pools and microbial extracellular enzyme activities [2]. These sites are characterized by a thin atmosphere, high UV radiation, extreme aridity, low year-round air temperatures and extreme diurnal soil temperature fluctuations [1,2,5]. The highest of the Atacama volcanoes is Volcán Llullaillaco, 6739 meters above sea level (m.a.s.l.), which is the location of the highest archaeological sites on Earth [6,7] and some of the best naturally preserved mummies anywhere in the world [8]. The lack of decay of these mummies suggests that this environment may be too dry and cold for the growth and activity of decaying microbes.

Initial studies of the biogeochemistry and microbiology of the high slopes of Volcán Llullaillaco and nearby Volcán Socompa showed that the soils on these volcanoes contained microbial communities of extremely low diversity, except in areas near fumaroles and penitentes [1,2,9,10] and had exceptionally low organic carbon content (0.005 to 0.017%) [2]. The functional nature of these communities remains debatable given the harshness of this environment and the low levels of biomass and diversity [11]. However, microbial communities have been shown to be active at sites that receive supplemental water and nutrients (from fumaroles and/or melting ice), even at elevations above 6000 m.a.s.l. [9].

High-elevation desert soils are also of great interest to the field of astrobiology, as the present climate of Mars can be compared with that of a very high mountain on Earth [12]. Today, the Martian surface is very cold, hyper-arid and experiences extreme thermal fluctuations. This was not always the case, as Mars orbiters and landers have obtained evidence that liquid water existed on the surface of Mars, possibly as recently as 300,000 years ago [13]. Despite the presence of water in the distant past, it is likely that any existing microbial life on Mars would have had to adapt to the conditions currently found on the surface. Extreme high-elevation systems, such as the high Andes, can therefore be a model system for potential life on Mars and their study can provide insights for the search of life on other habitable planets in our universe [4].

The true extent of the microbial growth and activity in extreme high-elevation environments is still unclear. Understanding factors that control the proportion of active microbes in the environment is critical to determine which microorganisms make a living in these harsh environments and are truly extremophiles and which are either dormant propagules from elsewhere or relic DNA [14]. Stochastic aeolian input from lower elevations results in the deposition of cells whose DNA can persist and be retrieved in 16S/18S rRNA gene surveys. However, it is likely that many of those microorganisms are neither surviving nor functioning in situ since they are not adapted to the harshness of the environment. It was generally assumed that the low diversity and biomass of extreme cryophilic environments was due to a lack of water [1,2,11,15,16,17]. However, recent studies have challenged this assumption by showing that nutrient additions to periglacial soils (>5000 m.a.s.l.) in the high Andes of Perú dramatically accelerated microbial community succession [18,19]. A few similar studies of soils of the Antarctic Dry Valleys also indicate that nutrients may limit microbial activity and diversity in cold–dry ecosystems. For instance, a natural gradient in water availability did not produce predictable responses in bacterial community characteristics across local and regional scales in the Taylor and Wright Valleys [20], while later mesocosm experiments showed that nutrient additions had a much stronger effect than water additions on microbial activity and community structure [21]. High elevation soils on Llullaillaco have some of the lowest nutrient levels ever measured in a terrestrial environment, which may indicate that these environments are more nutrient than water limited. It is possible that microbial activity can occur in this extreme environment during rare periods of transient water availability only if nutrients are available to support microbial growth.

A significant, but understudied, trait of this harsh high-elevation ecosystem is the daily fluctuation in temperature that drives diurnal freeze–thaw cycles [2,22]. The thin atmosphere, combined with intense solar radiation, creates freeze–thaw cycles that are among the most extreme yet observed for soils on Earth’s surface. Daily temperature cycling across the freezing point is a key challenge for microbial growth and survival at high elevations and high latitudes. Previous studies have shown that freeze–thaw cycle frequency influences microbial communities in the cryosphere [23] and limits net primary productivity in the extreme soils of the Dry Valleys of Antarctica, where daily fluctuations vary by more than 20 °C per day during the austral summer [16]. Mineral soils at a high elevation on Llullaillaco and other nearby volcanoes can experience even more extreme diurnal temperature fluctuations (>40 °C) than soils of the Antarctica Dry Valleys [2,24,25]. Recent work exposing soil microcosms to prolonged freeze–thaw cycles using a chamber that replicates field cycles (amplitude of 36 °C every 24 h) revealed that the original soil communities significantly shifted upon repeated freeze–thaw cycling [3]. This study also showed that the basidiomycete yeast *Naganishia friedmannii* (formerly *Cryptococcus friedmannii*), which is the dominant microeukaryote at high elevations on Volcán Llullaillaco, increased in relative abundance in soils subjected to freeze–thaw cycles when water was available and could grow relatively rapidly in pure culture during freeze–thaw cycles [3].

In the present study, we build on the previous work of Vimercati et al. (2016) [3] to characterize how microbial communities from arguably the harshest ecosystem on Earth respond under freeze–thaw stress when water and nutrient limitations are alleviated to gain insights into the taxonomic and genetic diversity of active microbes under realistic favorable conditions. Specifically, we carried out experiments where we amended high-elevation soils (>5100 m.a.s.l.) from Volcán Llullaillaco both in the field and in controlled microcosms in the lab to determine which community members respond to just water addition (simulating a snowmelt event) and those that respond to water and nutrients (simulating snowmelt combined with the presence of aeolian-deposited nutrients). 

We hypothesized that (1) based on our previous work, *Naganishia friedmannii* would increase in abundance as water and nutrients first become available, (2) the microbial community would undergo significant changes in richness and structure during freeze–thaw cycles when both water and water + nutrients become available, with the water + nutrient addition allowing the most pronounced change and (3) only a small fraction of the total community would be able to be active under realistic favorable environmental conditions.

## 2. Materials and Methods 

### 2.1. Soil Microcosm Manipulation

Soils collected at 5103 m.a.s.l. (8 March 2016) from the slopes of Volcán Llullaillaco (24°44.043, 68°34.573) were homogenized and added (13 g dry weight equivalent in each plate) to sterile microcosm plates (55 mm diameter Petri dishes, Fisher Scientific 8-757-13A, Waltham, MA, USA). Microcosms were incubated in a temperature-controlled chamber (Model TH024-LT, Darwin Chamber Co., Saint Louis, MO, USA) with a light–dark cycle of 12 h, and freeze–thaw temperature cycling (mean daily maximum = 26.42 (± 0.03 SE) °C, mean daily minimum = −10.07 (± 0.05) °C, Figure 1C and Appendix A) that mimics the thermal fluctuations experienced by field soils at a depth of 4 cm at high elevations on Llullaillaco [2,3]. Light in the chamber was supplied by full-spectrum fluorescent bulbs (Sylvania GRO-LUX^®^ F15T8/GRO/AQ/RP, Newhaven, UK and Philips F32T8/TL741, Amsterdam, Netherlands). Freeze–thaw cycles were programmed so that the rate of soil freezing did not exceed rates recorded near the field site from which the soils were collected [2,3,26].

Treatments were randomly assigned to each of the microcosm plates, yielding four replicates per treatment and controls. Treatments were (1) sterile deionized water addition (+W) to simulate occasional snowfalls (70% water holding capacity, 24–26% H_2_O) and (2) sterile deionized water + nutrient addition (+WN) to simulate aeolian deposition when water is available (inorganic nitrogen, phosphorus and organic carbon). Ammonium chloride (NH_4_Cl), monopotassium phosphate (KH_2_PO_4_) and sucrose were used as sources of N, P and C, at concentrations of 75 µg g^−1^, 75 µg g^−1^ and 100 µg g^−1^, respectively. The concentrations used were based on preliminary laboratory experiments that determined that these amounts ensure an adequate amount to overcome nutrient limitations without changing the ionic balance of the soil solution [27]. In addition, we tested the cumulative effects of repeated wet–dry cycles (as would occur on the mountain) by simulating three discrete snowfall episodes in both water addition (+W) and water + nutrient addition (+WN) microcosms. Note that +WN plots that received multiple water additions only received one nutrient addition. Controls consisted of microcosm plates (four replicates) that did not receive any nutrients or water but were incubated in the freeze–thaw chamber and were sampled after the 1st (C1) and 3rd (C3) snowfall events. There were no controls after the 2nd snowfall event because of the limited amount of soil available for the experiment. Four replicates of untreated microcosm soils that were kept at −20 °C were representative of the starting community. The arrangement of the plates in the incubator was randomized every 3 to 4 days to account for minor variations in temperature and light within the chamber.

The first subset of soil microcosms (four replicates per treatment and controls: C1, W1, WN1) was destructively sampled at 5 days after the first water and water + nutrient addition. The remaining soil microcosms were left in the chamber for an additional 6 days before receiving another water addition (2nd snowfall event) to restore the amount lost due to evaporation in the chamber after the first simulated snowfall event (up to 70% water holding capacity). Preliminary experiments have shown that approximately 90% of added water is lost after a total of 11 days in the chamber. Microcosm plates were sealed with parafilm (Bemis Company Inc, Neenah, WI, USA) after the 2nd and the 3rd water additions in order to decrease water evaporation rates. The second subset of soil microcosms (2nd snowfall event) was destructively sampled at 9 days after the second water addition (W2, WN2). The remaining soil microcosms were left in the chamber for an additional 13 days before receiving another water addition (3rd snowfall event) and were destructively sampled 9 days after the third water addition (C3, W3, WN3). To summarize, a total of 36 samples were included in the experiment: four starting soils, eight control soils, four +W (x3 simulated snowfall events) and four +WN (x3 simulated snowfall events).

### 2.2. Field Experiment

The field experiment was conducted in March 2016 at a site about a 100 m elevation above basecamp on the west-facing slope (5103 m.a.s.l., 24°44.043, 68°34.573) of Volcán Llullaillaco. Square plots of about 0.5 × 0.5 m were established, as shown in Figure 1, with each of the four replicate water addition plots paired with four replicate water + nutrient addition plots. The controls for each plot were soils from the plot sampled just before the water and nutrient amendment, such that each plot had a paired control sample prior to the amendment. Treatment plots received either (1) 500 mL of sterile deionized water (+W) or (2) 500 mL of sterile deionized water + nutrients (+WN). Nutrients were added in the form of ammonium chloride (NH_4_Cl), monopotassium phosphate (KH_2_PO_4_) and sucrose as sources of N (6 g/m^2^), P (5.8 g/m^2^) and C (12.2 g/m^2^), respectively. These concentrations were meant to simulate a pulse of nutrients that would significantly overcome any natural limitation. Plots amended with water and water + nutrients were sampled 6 days after the amendments. Conditions were cloudless during the experimental period and soil temperatures at a 4 cm depth cycled diurnally and ranged from +26.79 °C (± 1.77) to −2.60 °C (± 0.43) within 24 h (Appendix A).

### 2.3. DNA Extraction and Sequencing

Total DNA was extracted with a PowerSoil^®^ DNA Isolation Kit (MoBio Inc., Carlsbad, CA, USA), following the manufacturer’s instructions. For the amplification of the bacterial and archaeal 16S rRNA gene region, we used the oligonucleotide primer set 515F/806R [28], while for the amplification of the eukaryotic 18S rRNA gene, we used the Euk1391f/EukBr primer set (Earth Microbiome Project, accessible at http://www.earthmicrobiome.org/emp-standard-protocols/16S-18S/). Amplified DNA was pooled, normalized to equimolar concentrations using SequalPrep Normalization Plate Kits (Invitrogen Corp., Carlsbad, CA, USA), barcoded and then sequenced using the Illumina MiSeq V2 (2 × 150 bp chemistry) at the BioFrontiers Sequencing Core Facility at the University of Colorado at Boulder.

### 2.4. Plate Counts

To supplement the relative abundance data provided by DNA sequencing and verify the viability of cells, we also performed plate counts of colony-forming units (CFUs) of *N. friedmannii*. Wet soil (0.5 g) from each control and +WN replicate from the field experiment and from all replicates from the microcosm experiment was added to 5 mL of sterile water in 20 × 150 mm glass test tubes. The tubes were vortexed for 1 min, and 1 mL was pipetted into tubes containing 9 mL sterile water. This process was repeated twice more, such that each soil sample contained four tube dilutions. One hundred microliters were then spread evenly across Petri dishes containing an agar medium known to support *N. friedmannii*, as prepared by Vimercati et al. (2016). This resulted in final dilutions of 0.1, 0.01, 0.001 and 0.0001 per gram of wet soil. We took a second aliquot of 0.5 g from each sample to dry at 60 °C for 48 h to calculate the amount of dry soil that was diluted. Plates were incubated at 15 °C. We checked the plates and counted CFUs after 7 and 14 days. Only the dilution plate per sample containing between 30 and 300 colonies (or closest to this range) was counted. The most abundant colony types were viewed under a compound microscope to confirm colony identification as *N. friedmannii*. Two other microorganisms that also grew on plates were identified as actinomycetes and *Moesziomyces* (Appendix A).

### 2.5. Data Processing and Statistical Analysis

Sequences were demultiplexed, quality filtered and processed using the Quantitative Insights into Microbial Ecology (QIIME) v. 1.9.1 bioinformatics package [29]. The 16S rRNA gene paired-end reads were joined, but this process did not work for 18S rRNA gene reads due to insufficient overlap, so only the read corresponding to the 1391F primer was used. This read was selected because it overlapped more with most sequences within the NCBI and SILVA databases [30]. Singletons were excluded from further analysis and sequences with >97% identity were clustered into operational taxonomic units (OTUs) via UCLUST [31]. Representative sequences for each OTU were chosen for taxonomic classification with the Greengenes 13.5 [32] and Silva Ref NR 97 databases [33] for 16S and 18S rRNA gene sequences, respectively. All mitochondrial and chloroplast OTUs based on this classification were removed from the bacterial data set and all bacterial OTUs were removed from the eukaryotic data set. The taxonomic assignments of the top OTUs from each treatment were verified by using BLAST (NCBI online tool) and refined as needed. Sequences were aligned with PyNAST [34] and a phylogeny was built with the FastTree algorithm [35]. OTU tables were rarified to the number of sequences in the lowest populated sample to make comparisons more robust and were used to assess the alpha diversity and relative abundance of all taxa. The richness of OTUs was estimated for the rarefied 16S and 18S rRNA gene datasets in QIIME. For the microcosm experiment, we analyzed differences in OTU richness with a two-way analysis of variance (ANOVA) with the R function “aov”, including an interaction term, to test if water and/or water + nutrient additions significantly affected richness after multiple simulated snowfall events. We used Tukey’s honest significant differences (HSD) for post hoc pairwise comparisons. Colony-forming unit abundances were analyzed with either ANOVA on log-transformed data followed by Tukey’s post hoc test (for *N. friedmannii* colonies) or zero-inflated models with a Poisson distribution followed by Tukey’s post hoc test (for actinomycete and *Moesziomyces* sp. colonies, implemented with the “pscl” and “emmeans” R packages, Jackman 2017, Lenth 2019). For the field experiment, *N. friedmannii* relative abundances and colony-forming unit abundances were analyzed with paired *t*-tests (paired control and water addition plots, paired control and water + nutrient addition plots, paired water addition versus water + nutrient addition plots). A community-level distance matrix with pairwise weighted UniFrac beta-diversity values was generated and analyzed with permutational multivariate analysis of variance (PERMANOVA) corrected for multiple comparisons using the “adonis” function [36] to partition the variance in community composition attributable to water versus water + nutrients addition. Post hoc pairwise comparisons were generated with the “RVAideMemoire” R package [37] to ask how assemblage structures differed among treatments. We tested the homogeneity of multivariate dispersion (PERMDISP) with the “betadisper” function to evaluate whether the difference supported by PERMANOVA was due to differences between clusters or differences between the variations within clusters. Principal coordinate analysis (PCoA) ordination was constructed based on weighted UniFrac distance matrices in order to visualize differences among community compositions of treatments. Similarity percentage analysis (SIMPER) was used to determine the OTUs that contributed most to the observed dissimilarity among treatments. All statistical tests, unless otherwise stated, were performed using the “vegan” R package [36], in R v. 3.6.1 [38].

## 3. Results

### 3.1. Naganishia friedmannii Response

Because our previous work showed that *N. friedmannii* increased in relative abundance during 2 months of freeze–thaw cycles when liquid water was available [3], we wished to determine if this organism could respond on a shorter (more realistic to field scenarios) time frame (less than a week). Liquid water following snowmelt is likely available only for a limited amount of time, on the order of days, rather than months, due to high sublimation rates in the environment (Schubert 2014). *N. friedmannii* (100% match to AB032630) was the most abundant OTU in the starting soils of both the field and microcosm experiments (Figure 2) and significantly increased in relative abundance and colony-forming units after only 6 days in the +WN microcosm (Tukey’s HSD *p* < 0.01), but not in the microcosm receiving only water (Figure 3). *N. friedmannii* also increased in the field experiment after only 6 days in the +WN plots compared to the starting soil, but this increase was not significant (paired *t*-test *p* = 0.12). There was also no significant increase in colony-forming units in the field experiment (paired *t*-test, *p* = 0.33). The difference in the number of CFU counts of *N. friedmannii* between the microcosm and field experiment could be explained by their different rates of water evaporation, influencing overall water availability. In the microcosms, *N. friedmannii* decreased in relative abundance on subsequent water additions and in fact decreased significantly compared to the control (C3 vs. W3, Tukey’s HSD *p* = 0.003) in microcosms receiving only water after the third addition (Figure 2). However, plate counts demonstrated that the number of colony-forming units continued to increase in the second and third water and nutrient additions, with samples receiving three additions having significantly higher CFU counts than those receiving only one addition (Tukey’s HSD *p* < 0.05, Appendix A). This suggests that the decreases seen in the relative abundance data are due to increases in other organisms, not true decreases in *N. friedmannii*. Furthermore, CFU counts were significantly higher in all + WN microcosms receiving additions of water compared to the starting soils and controls (Tukey’s HSD *p* < 0.05, Appendix A).

### 3.2. Alpha and Beta Diversity Patterns of Soil Microcosms

Alpha and beta diversity patterns were significantly affected by both water and water + nutrient additions under freeze–thaw cycles for both bacterial (16S) and eukaryotic (18S) communities. The 16S OTU richness (alpha diversity) was significantly affected by the different treatments (ANOVA *p* < 0.001, *F* = 7.5) (Figure 4a). More specifically, post hoc comparisons showed that both +W3 and +WN3 16S OTU richness were significantly lower than that of starting soil (Tukey’s HSD *p* < 0.02; *p* < 0.001, respectively) and control 3 microcosm soils (*p* < 0.01; *p* < 0.001, respectively) (Figure 4a). The 18S OTU richness was also significantly affected by the different treatments (ANOVA *p* < 0.001, *F* = 19.1) (Figure 4b). Specifically, post hoc comparisons showed that OTU richness of all +W and +WN treatments was significantly lower than that of both starting and control 3 microcosm soils (Tukey’s HSD *p* < 0.01) (Figure 4b).

The analysis of both 16S and 18S beta diversity showed the clustering of communities based on treatment (PERMANOVA *p* < 0.001, *R^2^* = 0.75; *p* < 0.001, *R^2^* = 0.77; respectively). More specifically, bacterial assemblages of all treatments were significantly different from each other except for starting and control microcosm soils, +W2 and +W3, and +WN1 and +WN2. Eukaryotic microbial assemblages of all treatments were significantly different from each other except for +W2 and +WN3. The 16S PCoA ordination revealed that microbial communities were increasingly different from both starting and control soil microcosms with each water addition in both treatments (Figure 5). The 18S PCoA ordination also showed a similar pattern (Figure 6). The microbial communities of starting soil microcosms clustered separately from those of control soil microcosms, showing that thermal fluctuation by itself affects the community (PERMANOVA *p* < 0.05). The test for the homogeneity of dispersion was not significant for 16S communities (PERMDISP *p* = 0.078, *F* = 1.92), showing that the main factor driving community clustering was the difference between treatments rather than within treatments. On the other hand, 18S communities from different treatments displayed significantly different within-treatment variance (PERMDISP *p* < 0.05, *F* = 8.26).

### 3.3. Taxonomic Shifts of Soil Microcosms

#### 3.3.1. Bacterial Community Response

Bacterial sequences in all treatments were dominated by six phyla: Actinobacteria, Proteobacteria, Bacteroidetes, Chloroflexi, Verrucomicrobia and Acidobacteria (Appendix A). The mean dissimilarity of the bacterial communities across all treatments was 35% (SIMPER analysis, Table 1). Pairwise comparisons between treatments had a mean dissimilarity range of 28–46%. Five OTUs comprising 39% of the total relative abundance accounted for most of the community compositional differences between treatments. The top five OTUs that explained the most variance between each treatment pair were identified as members of the Oxalobacteraceae (Betaproteobacteria), Chitinophagaceae (Bacteroidetes), Solirubrobacterales (Actinobacteria), Pseudonocardiaceae (Actinobacteria) and Chthoniobacteraceae (Verrucomicrobia) (Figure 5). The Oxalobacteraceae OTU, whose closest match in the NCBI database was a *Noviherbaspirillum* sp. (KM260006, 100% identity), significantly increased in relative abundance to 9.4% on the third +W treatment compared to both control (0.7%) and starting (1.4%) microcosms (Tukey’s HSD *p* < 0.05). The Chitinophagaceae OTU increased in relative abundance in +W (12.1%) and +WN (9.7%) treatments compared to starting soil microcosms (8.4%). The Solirubrobacterales OTU had the greatest increase in relative abundance in +WN treatments (13.6%) compared to starting soil microcosms (11.1%); however, they decreased in abundance with multiple water additions in +W treatments (9.2 to 5.1%). Pseudonocardiaceae declined in relative abundance with multiple water additions in both +W (4.2 to 0.9%) and +WN (2.9 to 1%) treatments compared to both starting (5.2%) and control (5.2%) soil microcosms. A number of additional OTUs explained the variance among multiple water additions in the two treatments of this study: Chloracidobacteria significantly increased after the first water addition to the +WN1 treatment (Tukey’s HSD *p* < 0.05); Chloroflexi significantly declined with multiple water additions in +W treatments (Tukey’s HSD *p* < 0.05); Actinobacteria within the *Phycicoccus* genus significantly increased (Tukey’s HSD *p* < 0.05) in the +WN3 treatment.

#### 3.3.2. Eukaryotic Community Response

There was a profound shift in the eukaryotic community in both +W and +WN treatments (Figure 2). The mean dissimilarity of eukaryotic communities across all treatments was 60% (SIMPER analysis, Table 2) and pairwise comparisons between treatments had a mean dissimilarity range of 52–67%. Six OTUs, comprising 74% of the total relative abundance, were the main drivers of the compositional difference among treatments.

*Naganishia friedmannii*’s response to multiple water and water + nutrient additions was described above. Multiple water additions to +W treatments resulted in a significant increase in the fungal species *Moesziomyces antarcticus* (MH188429, 100%), whose relative abundance significantly increased (Tukey’s HSD *p* < 0.05) in +W2 (29%) and +W3 (41%) treatments. This result was corroborated by the growth of white fungal colonies, likely to be Moesziomyces, only on plates receiving multiple water additions (+W2, +W3, +WN3), with the greatest number of colonies in the +W3 treatment (Tukey’s HSD, *p* < 0.05, Appendix A). By the third water addition, +W3 samples showed a significant increase (Tukey’s HSD *p* < 0.05) in the relative abundance of the Chlorophyta *Neochlorosarcina negevensis* (MG022670, 100% identity). A fungal OTU within the Dothideomycetes class significantly (Tukey’s HSD *p* < 0.05) increased in abundance with multiple water additions in +WN samples (5.3% in +WN1, 15.4% in +WN2, 17.2% in +WN3). This OTU is a 100% match with *Alternaria atra* (MH864090; [39]). Cercozoa within the Thecofilosea class significantly declined (Tukey’s HSD *p* < 0.05) in all treatments. Two green algae OTUs within the Chlamydomonadales order were only detected in starting microcosm soils (5.7 and 8.5%) but were absent from all treatments. The first of these OTUs was only distantly related to the genus Ploeotila (96.8% identity, HQ404867), while the second OTU was a 100% match for a *Chlamydomonas* sp. (LC380251, [40]).

### 3.4. Environmental Community Response to Water and Nutrient Additions

In the field experiment, a single application of water and water + nutrients did not result in a significant shift in the overall community composition for either bacteria or eukaryotes (PERMANOVA, *p* > 0.05). However, as discussed above, there was a trend of an increase in the relative abundance of *Naganishia friedmannii* in the water + nutrient treatment (paired *t*-test, *p* = 0.12). In the bacterial community, an Oxalobacteraceae OTU showed a significant increase in relative abundance in the water treatment (0.5 to 1.1%, paired *t*-test *p* = 0.048). This was the same *Noviherbaspirillum* OTU that increased in relative abundance after multiple water additions in the microcosm experiment.

## 4. Discussion

In order to gain insight into how the microbial communities of high elevation soils of the Atacama respond to water and nutrient availability under environmentally relevant thermal fluctuations, we performed manipulative experiments using tephra soils collected at 5100 m.a.s.l. on Volcán Llullaillaco. While our previous work showed that these sites are too dry to support microbial activity at ambient water availabilities (0.25% water) [2,4], our work here addressed the question of whether members of the microbial community could respond during extreme freeze–thaw cycles to the increases in water availability that occur after rare snowfall events and with and without nutrient addition.

Our experiments show that both bacterial and eukaryotic communities were significantly affected by water and water + nutrient additions in microcosms under field thermal conditions, which supports our second hypothesis (Figure 5 and Figure 6). Given the extremely low abundance of nutrients in this soil, we predicted a greater response when both water and nutrients become available, but our experiment also shows that a significant response is detected when just water becomes available. Water alone, independent of nutrient availability, is therefore a key abiotic factor driving the response of the microbial community, as previously shown in the same environment but at a higher elevation [3]. Microorganisms inhabiting these soils may be able to take advantage of the low nutrients found in this environment as soon as there is water, which could explain why there is an effect on the community just by water addition. We observed that a single application of water or water + nutrients was enough to cause a significant change in both bacterial and eukaryotic community structure. This result suggests that these communities, which have to rely on stochastic inputs of water and nutrients in their environment, have adapted to quickly respond when resources are available.

The overall community shifts seen in the microcosms were not seen in the field, which may be explained by the communities not having enough time to actively respond to the newly available resources. Cloudless conditions and extreme aridity may have caused the amended water to evaporate too fast for the community to take advantage of the water and nutrient pulse. Snowfalls are known to occur on Llullaillaco, but little is known about the persistence of snow cover and it is possible that soil communities have water availability for a longer amount of time than that provided by our environmental manipulation.

### 4.1. Naganishia friedmannii Response in Microcosms and in the Field

An OTU closely related to the basidiomycete yeast *Naganishia friedmannii* showed an increase in relative abundance and colony-forming units after the first amendment of water and especially water + nutrients to the soil microcosms, which supports our first hypothesis. This *Naganishia* sp. is most closely related to the endolithic and xerotolerant *Naganishia friedmannii* isolated from the Dry Valleys of Antarctica and high elevation sites of the Himalayas [2,3]. *N. friedmannii* is abundant in polar and high-elevation environments [41] and it has been reported to exhibit polyextremophilic aptitudes [42] that range from high resistance to dehydration–rehydration [43], extreme temperature fluctuations [3], exposure to UV, desiccation and stratospheric conditions [44]. Its ubiquity in extreme high-elevation and high-latitude soils and its ability to withstand and grow under multiple stressors suggest that *N. friedmannii* may be an important functional component of these extreme environments. Our soil microcosm experiment showed that *N. friedmannii* was the first microeukaryote to take advantage of the water + nutrient additions, further lending support to the idea that this extremophilic yeast may be intermittently metabolically active in this “barren” landscape. Our working hypothesis is that *N. friedmannii* is a versatile opportunist, able to persist in these extreme soils by increasing in numbers during rare times of water and substrate availability and possibly entering long periods of dormancy in between such events [3,26].

### 4.2. Bacterial Community Response in Soil Microcosms

#### 4.2.1. Oxalobacteraceae

Among the bacterial taxa that were mostly responsible for the community shifts, *Noviherbaspirillum* (Oxalobacteraceae) significantly increased in soil microcosms with multiple water additions (but not when nutrients were also added). This OTU also was the only bacterial OTU that increased significantly in the field water addition plots after only six days. Together, these data indicate that this organism may be uniquely adapted to responding to temporary water inputs even without added nutrients, making it a good candidate for future studies of life in these extremely oligotrophic soils. The Oxalobacteraceae family is widespread in cryophilic and oligotrophic environments, such as glacier-fed streams [45], glacier forefields [46], cryoconites [47] and high-elevation periglacial soils [48]. This family is metabolically diverse, and some genera are adapted to oligotrophic conditions [49], which may allow them to use different aeolian-deposited carbon sources. Some members of the genus *Noviherbaspirillum* have also been shown to be resistant to gamma radiation [50] and close relatives of this OTU carry the *nif* genes [49], which indicates that this OTU may make nitrogen available in this nutrient-limited environment. Their lower relative abundance in the soil microcosms that received both nutrient and water additions adds evidence for this group being oligotrophic and therefore outcompeted by different phylotypes that show a faster response to carbon and nutrient additions.

#### 4.2.2. Bacterial Phototrophs

Cyanobacteria, which are found in a very low abundance in these soils, did not respond to water amendments in either the field or microcosm experiments. Cyanobacteria have been found in the hyper-arid low-elevation soils of the Atacama [51] and are associated with water sources, such as fumaroles and penitentes, at high elevations on Atacama volcanoes [1,9,10], but they are undetectable in the most extreme higher-elevation sites (5300 to 6300 m.a.s.l.) on the volcanoes Llullaillaco and Socompa [2,9]. Previous research has shown that cyanobacteria are relatively rare in melting snowfields and other non-stable cryosystems [52,53]. Therefore, they may not even be transiently active in the soils studied here but may instead be dormant propagules blown in from other environments.

### 4.3. Eukaryotic Community Response in Soil Microcosms

In addition to the response of *N. friedmannii* discussed previously, several other eukaryotes responded significantly to the resource amendments, especially after multiple treatments. A summary of these responses and information about these organisms are presented in Table 3.

#### 4.3.1. Dothideomycetes

Fungi within the Dothideomycetes class significantly increased in relative abundance with multiple water additions in soil microcosms that were amended with nutrients. Many Dothideomycetes are known to be resistant to a number of extreme environmental conditions by developing avoidance strategies [54]. For example, they can tolerate oligotrophic conditions, repeated freeze–thaw stress [55,56], radiation and dehydration [57] by producing extracellular polymeric substances (EPSs), melanin and compatible solutes. Many genera can form deeply melanized, basally meristematic colonies and produce extracellular polysaccharides to ameliorate UV and osmotic stresses [58]. Their compact colony morphology helps minimize exposure to external stressors by decreasing their surface to volume ratio and appears to have optimized a slow but persistent growth strategy for extremely cold and barren habitats [59]. They are characterized by simple life cycles that can be completed during short periods of time when favorable conditions prevail [54]. The oligotrophy of the extreme environments where they are found and the high metabolic costs to synthetizing compounds related to stress resistance account for their slow growth velocity [60] and help explain why the relative abundance of this taxon increases significantly only after the second and third water additions to the +WN treatments in this experiment.

#### 4.3.2. *Moesziomyces* sp.

An OTU closely related to the fungal species *Moesziomyces antarcticus* (formerly *Candida antarctica*) significantly increased in relative abundance with multiple water additions to soil microcosms, with the highest increase after the third water addition (+W3). This yeast was originally isolated from Lake Vanda in the Wright Valley, Antarctica [61] and produces cold-active enzymes and biosurfactants [62]. Biosurfactants have been shown to have different functions, such as anti-agglomeration effects on ice particles [62]. Biosurfactants can also be part of cell envelopes and participate, together with other components such as exopolysaccharides, in protection against high salinity, temperature and osmotic stress [63]. Our study provides new evidence that, along with its ability to withstand cold temperatures through cold-adapted enzymes and other molecules, this yeast is also able to increase in relative abundance under extreme thermal fluctuations and very low nutrient contents.

#### 4.3.3. *Neochlorosarcina* sp.

By the third water addition to +W microcosms, we observed a significant increase in the relative abundance of an OTU that was a 100% match with *Neochlorosarcina negevensis,* a member of the Chlorophyceae. This alga is commonly found in biological soil crusts of drylands [64]. Layers of algae within the Chlorophyceae have been identified as being the closest phototrophs to the soil surface within gypsum deposits at low elevations in the Atacama Desert [51]. Many Chlorophyceae have the capability to synthesize very high amounts of complex molecules, such as carotenoids and EPSs, that enhance cell resistance under unfavorable environmental conditions, including excess light and UV, water and nutrition stress [65,66,67]. Other strategies for survival in harsh conditions include the development of dormant stages or avoidance through motility [68]. Previous work has shown that in arid environments, high relative humidity is enough to induce the metabolic activity of some algae [69]; however, more work will be needed to determine if *Neochlorosarcina* sp. is a functional phototroph in this environment. Of broader significance, this is the first report of phototrophs on the high elevation slopes of Volcán Llullaillaco not associated with semi-permanent water sources such as fumaroles or penitentes [1,9,10]. This ecosystem was previously thought to be completely heterotrophic- or chemolithotrophic-dependent [11], although there is evidence that photoheterotroph bacteria in the Rhodospirillales are present at the highest sites yet sampled on Llullaillaco [2,70]. However, it is important to note that the present study was conducted using soils from 5100 m.a.s.l., whereas the studies mentioned above were conducted on soils from between 5300 and 6300 m.a.s.l. Therefore, more work is needed to determine the elevational limit to phototrophic eukaryotic life not associated with penitentes or fumaroles on these volcanoes.

Together, the results from the bacterial and eukaryote communities support our third hypothesis that only a small fraction of the community is active under favorable environmental conditions. Interestingly, there were more eukaryotic taxa than bacterial taxa that responded to the resource amendments, suggesting that this may be a eukaryotic-dominated system.

### 4.4. Astrobiological Implications

This work has implications for the prospect of life on Mars, which transitioned from an earlier wetter environment to today’s extreme hyper-aridity [71]. Recent studies have demonstrated that microbial activity can be sustained even in the driest soils of the lower Atacama Desert for very long periods of time following an episodic increase in moisture [72]. Our findings expand the range of hyper-arid and cold environments temporarily habitable for terrestrial life, which by extension applies to other planetary bodies, such as Mars. This transitory habitat with microorganisms active for short periods of time at the highest elevations of the Atacama can serve as a reasonable working model for Mars.

## 5. Conclusions

Life is nearly ubiquitous on Earth and microbial communities have been found in some of the harshest ecosystems. While 16S and 18S rRNA gene surveys have revealed their presence, the activity and metabolic status of microbial communities in these harsh environments remains poorly understood. Experimental manipulation is an important tool to reveal the potential of these communities and evaluate their response to less restricting conditions. We studied the response of a highly specialized extremophilic community from the high elevations of an Atacama volcano to the episodic availability of water and water coupled with nutrients. Our study suggests that occasional snowfall events, coupled with nutrients provided by aeolian deposition, can provide a habitable environment for microorganisms that can become metabolically active in this harsh ecosystem. Along with a 16S and 18S rRNA survey, our investigation provided new insights into the question of which microbial extremophiles would potentially be active just in response to increased water and which need the additional input of nutrients. Only a small number of taxonomic groups drove the community shift and they likely reflect the active fraction of the community under favorable conditions. Determining the physiological state of microbial cells found in extremely dry and temperature-fluctuating environments, specifically whether microorganisms are dormant or metabolically active, is of fundamental importance for the NASA Astrobiology Roadmap [73] that aims to characterize the boundaries of life in putative Earth analogues, and in particular the dynamics of organism survival and reproduction in conditions currently present on the surface of Mars.

## Figures and Tables

**Figure 1 microorganisms-08-01061-f001:**
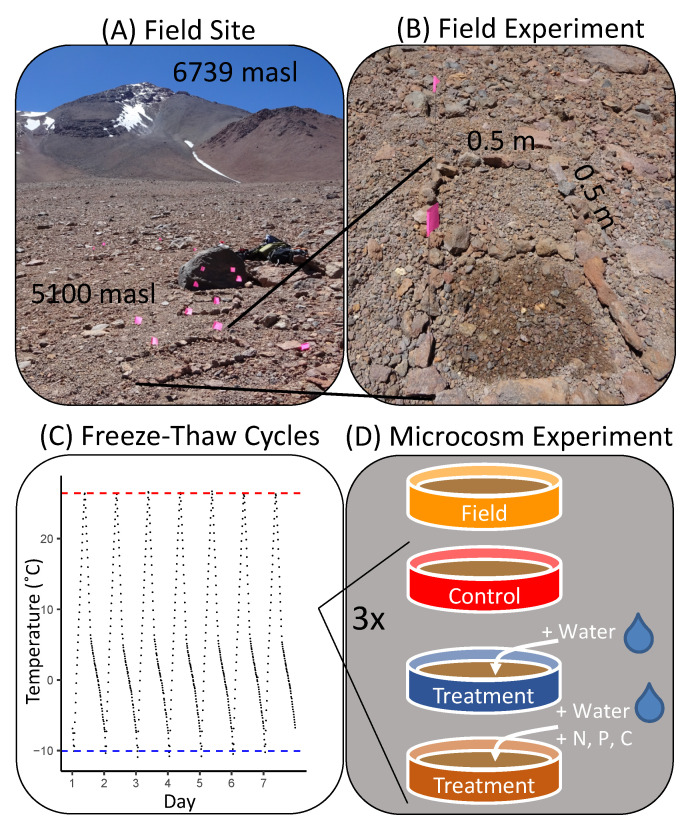
Study site and experimental design. (**A**) Location of soil sampling site at 5103 m.a.s.l. on Volcán Llullaillaco (24°44.043, 68°34.573). One of the false summits (~6400 meters above sea level (m.a.s.l).) of Volcán Llullaillaco is visible in the background. (**B**) Close-up view of one of the plots of the environmental manipulation experiment: the top plot has not been amended yet, while the bottom plot received water addition. (**C**) Temperature cycling as recorded in the environmental chamber over the course of the microcosm experiment. The red dashed line represents the mean daily maximum temperature of 26.42 °C. The blue dashed line represents the mean daily minimum temperature of −10.07 °C. (**D**) Treatments of the soil microcosm experiment: initial field soil and controls, water additions, and water and nutrient additions exposed to freeze–thaw cycles, each replicated four times.

**Figure 2 microorganisms-08-01061-f002:**
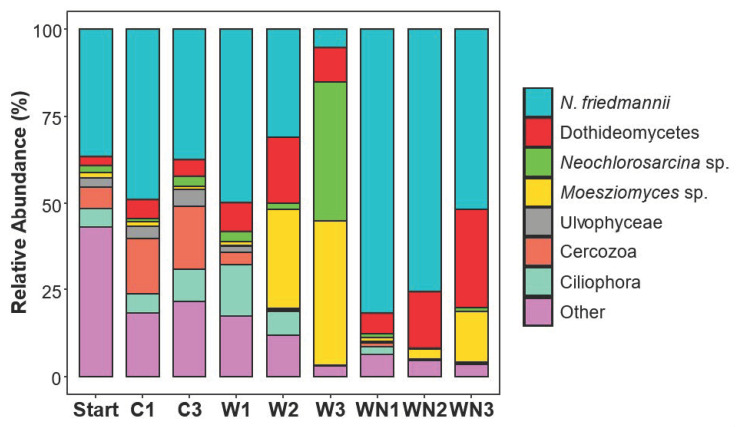
Taxonomy of the eukaryotic community shift in response to water (+W) and water + nutrients (+WN) amendments in the microcosm experiment. Stacked bar graphs showing relative abundances of dominant eukaryotic taxa (from 18S rRNA gene sequencing data). Taxonomy is from the Greengenes database. Bars represent means (*n* = 4). Most of the “Other” category in the starting soil is plant material (Vimercati et al. 2016). A stacked bar graph of bacterial community shifts can be found in Appendix A. *N. friedmannii*, *Neochlorosarcina* sp., *Moesziomyces* sp. and Ulvophyceae are each represented by one OTU. Dothideomycetes are all in the Pleosporales order.

**Figure 3 microorganisms-08-01061-f003:**
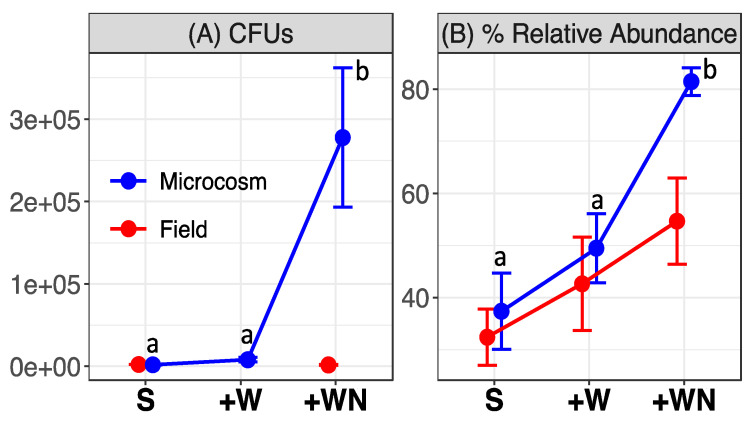
Mean (± SE) of the (**A**) number of colony-forming units per gram of dry soil of *N. friedmannii* and (**B**) relative abundance of *N. friedmannii* among the treatments in the microcosms (after first water addition and first water + nutrient addition) and in the field (samples collected 6 days after water and water + nutrient additions). The number of colony-forming units and the relative abundance of *N. friedmannii* increased significantly in the water + nutrient treatment in the microcosms (Tukey’s honestly significant difference (HSD) *p* < 0.05). In the field experiment, relative abundance increased slightly but not significantly in the +WN treatment (paired *t*-test *p* = 0.12), while colony-forming units did not increase significantly (paired *t*-test *p* = 0.33). S = starting soil (control), +W = water addition (only the first water addition in microcosms), +WN = water and nutrient addition (only the first water addition in microcosms). Note the difference in the *y*-axis scale between the two panels.

**Figure 4 microorganisms-08-01061-f004:**
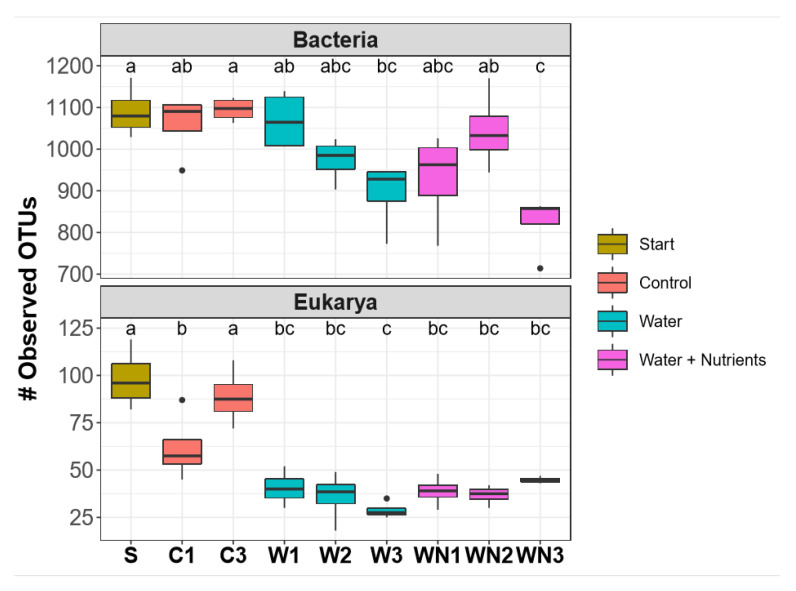
Alpha diversity (Operational Taxonomic Unit (OTU) richness) box plots for eukaryotic and bacterial communities in the microcosm experiment. Upper Panel: Bacterial community. Significant decrease in alpha diversity after 3rd water addition (Tukey’s HSD *p* < 0.05). Lower Panel: Eukaryotic community. Significant decrease in alpha diversity after 3rd water addition and in all water + nutrient additions (Tukey’s HSD *p* < 0.05). The dark horizontal line inside the box represents the median. The long vertical black lines for each box represent the maximum and minimum values of OTU richness and circles represent outliers. Boxes with same letter are not significantly different. Different letters represent significant differences among treatments (Tukey’s HSD *p* < 0.05). Note the difference in the *y*-axis scale between the two panels.

**Figure 5 microorganisms-08-01061-f005:**
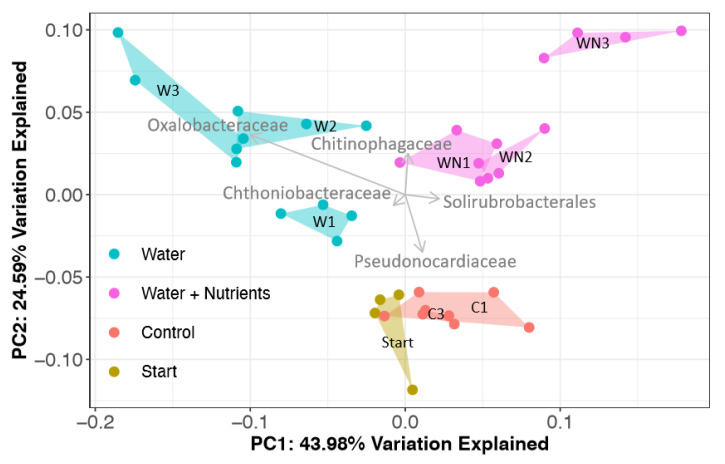
Principal coordinate analysis (PCoA) ordination plot of bacterial communities in soil microcosms. Repeated water additions to water- and nutrient-amended microcosms exposed to freeze–thaw cycles significantly changed the bacterial community (Permutational Multivariate Analysis of Variance (PERMANOVA) *p* < 0.001, R^2^ = 0.74). The vector length is proportional to the correlation between the PCoA axes and the OTUs that contributed most to the observed dissimilarity among treatments (Similarity Percentage analysis, SIMPER). Communities became increasingly different from controls following multiple amendments.

**Figure 6 microorganisms-08-01061-f006:**
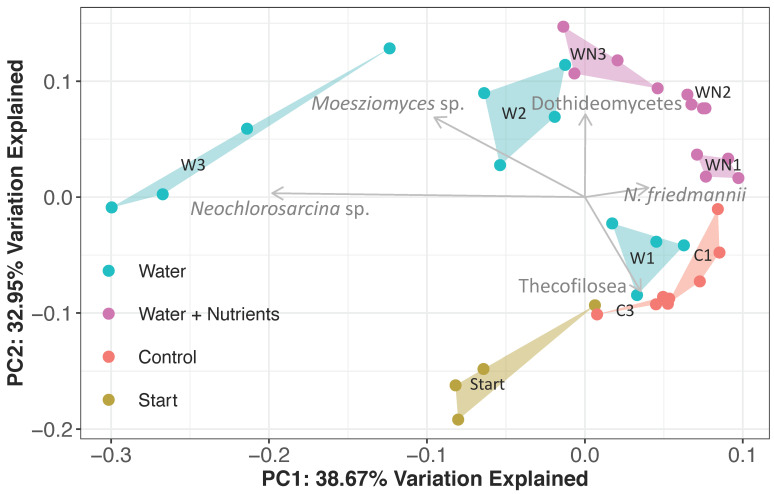
Principal coordinate analysis (PCoA) ordination plot of eukaryotic communities in soil microcosms. Repeated water additions to water- and nutrient-amended microcosms exposed to freeze–thaw cycles significantly changed the eukaryotic community (PERMANOVA *p* < 0.001, R^2^ = 0.75). The vector length is proportional to the correlation between the PCoA axes and the OTUs that contributed most to the observed dissimilarity among treatments (SIMPER). Communities became increasingly different from controls following multiple amendments.

**Table 1 microorganisms-08-01061-t001:** Similarity percentage analysis (SIMPER) among microcosm treatments for 16S data. For each comparison, the unique OTU identifier, phylum, family, percent contribution to dissimilarity and cumulative percent contribution to dissimilarity are shown for the top three taxa contributing to the dissimilarity.

Comparison	OTU ID	Phylum	Family	%	Cumulative
				Contribution	%
Start—Water	denovo6723	Betaproteobacteria	Oxalobacteraceae	7.08	7.08
Mean = 35%	denovo7247	Actinobacteria	Pseudonocardiaceae	3.58	10.66
	denovo3383	Actinobacteria	Solirubrobacterales	2.82	13.48
Control—Water	denovo6723	Betaproteobacteria	Oxalobacteraceae	6.82	6.82
Mean = 36%	denovo3383	Actinobacteria	Solirubrobacterales	3.87	10.69
	denovo7247	Actinobacteria	Pseudonocardiaceae	3.42	14.11
Start—Water + Nutrients	denovo7247	Actinobacteria	Pseudonocardiaceae	3.57	3.57
Mean = 36%	denovo11840	Verrucomicrobia	Chthoniobacteraceae	3.00	6.56
	denovo8881	Actinobacteria	*Phycicoccus*	2.65	9.22
Control—Water + Nutrients	denovo7247	Actinobacteria	Pseudonocardiaceae	3.56	3.56
Mean = 35%	denovo7213	Bacteroidetes	Chitinophagaceae	2.89	6.45
	denovo15971	Bacteroidetes	Chitinophagaceae	2.86	9.31
Water–Water + Nutrients	denovo6723	Betaproteobacteria	Oxalobacteraceae	7.01	7.01
Mean = 37%	denovo3383	Actinobacteria	Solirubrobacterales	4.72	11.73
	denovo11840	Verrucomicrobia	Chthoniobacteraceae	2.55	14.28

Mean dissimilarity among samples = #%.

**Table 2 microorganisms-08-01061-t002:** Similarity percentage analysis (SIMPER) among microcosm treatments for 18S data. For each comparison, the unique OTU identifier, phylum, family, percent contribution to dissimilarity and cumulative percent contribution to dissimilarity are shown for the top three taxa contributing to the dissimilarity.

Comparison	OTU ID	Phylum	Closest Taxonomy	%	Cumulative
				Contribution	%
Start—Water	denovo259	Basidiomycota	*Moeszyomyces* sp.	16.76	16.76
Mean = 68%	denovo1264	Basidiomycota	*Naganishia friedmannii*	15.27	32.03
	denovo251	Chlorophyta	*Neochlorosarcina* sp.	10.29	42.32
Control—Water	denovo259	Basidiomycota	*Moeszyomyces* sp.	18.29	18.29
Mean = 63%	denovo1264	Basidiomycota	*Naganishia friedmannii*	17.16	35.45
	denovo251	Chlorophyta	*Neochlorosarcina* sp.	11.39	46.84
Start—Water + Nutrients	denovo1264	Basidiomycota	*Naganishia friedmannii*	29.33	29.33
Mean = 59%	denovo1929	Dothideomycetes	Pleosporales (order)	9.90	39.23
	denovo264	Chlorophyta	*Chlamydomonas* sp.	6.11	45.34
Control—Water + Nutrients	denovo1264	Basidiomycota	*Naganishia friedmannii*	28.04	28.04
Mean = 52%	denovo1929	Dothideomycetes	Pleosporales (order)	10.27	38.31
	denovo598	Cercozoa	*Thecophilosea* sp.	9.53	47.84
Water–Water + Nutrients	denovo1264	Basidiomycota	*Naganishia friedmannii*	36.05	36.05
Mean = 59%	denovo259	Basidiomycota	*Moeszyomyces* sp.	17.79	53.84
	denovo251	Chlorophyta	*Neochlorosarcina* sp.	12.24	66.08

Mean dissimilarity among samples = #%.

**Table 3 microorganisms-08-01061-t003:** Summary of key eukaryotic taxa, their responses to water and nutrient additions, their rate of response (fast = responded after first addition; slow = responded after multiple additions), their potential general evolutionary strategy and their mechanisms of stress resistance from the literature.

Taxon	Organism Type	+ Water	+ Water + Nutrients	Increase Rate	Strategy	Mechanisms of Stress Resistance
*N. friedmannii*	Basidiomycete yeast	Neutral	Increase	Fast	r-selected	Dormancy
*Moesziomyces* sp.	Basidiomycete fungus	Increase	Increase	Slow	K-selected	Cold-active enzymes, biosurfactants
Dothideomycetes	Ascomycete fungus	Increase	Increase	Slow	K-selected	Extracellular polymeric substances, melanin, compatible solutes
*Neochlorosarcina* sp.	Alga	Increase	Neutral	Slow	K-selected	Extracellular polymeric substances, carotenoids, dormancy, motility

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
