# Peer review of "Limited Response of Indigenous Microbes to Water and Nutrient Pulses in High-Elevation Atacama Soils: Implications for the Cold–Dry Limits of Life on Earth"

_microorganisms, 2020, doi:10.3390/microorganisms8071061_

Round 1
Reviewer 1 Report
The submitted manuscript studied the response of indigenous microbes from the high elevations and hostile environment of Atacama volcano to episodic availability of water and water coupled with nutrients. The present study is based on the previous work of authors (Vimercati et al., 2016). In the Introduction, the authors proposed three hypotheses that were confirmed experimentally. The main conclusion of the manuscript is that occasional snowfall events coupled with nutrients provided by Aeolian deposition can provide a habitable environment for microorganisms that can become metabolically active in this harsh environment. The experiments were carried out both in the field and in the lab to investigate the response of the microbes to water and nutrition supplementation. In general, all parts (Introduction, Methods, Results, Discussion, and Conclusions) are described sufficiently and rigorously. Based on the described methodology, it is possible to repeat the experiments. Statistical tools are used correctly. Also, data interpretation is clear and leads to reasonable conclusions. The English language and style are fine. However, I have some concerns regarding the design of the experiments (please see the questions below). It is a pity that only a single application of water and water + nutrients were performed in the field experiment, so only a small part of the results come from a real field environment. From this point of view, the Title and the Abstract are a little bit confusing. The design of experiments in the laboratory, which should realistically simulate the in-situ conditions, is, in my opinion, the main limitation of the manuscript (please see the questions below). On the other hand, the main strength of the manuscript is in data processing and interpretation. In conclusion, after reasonable answering of the attached questions, I would endorse the publication of the manuscript with minor revisions.
My comments and questions:
Title and Abstract: The Title and the Abstract should contain information that most of the results come from laboratory experiments, not from the real (field) environment.
Abstract, Lines 27-29: “These results indicate that only a fraction of the detected community is able to become active under realistic favorable conditions, and that water alone can dramatically change community structure.” To make such a (realistic) conclusion, it is necessary to show that your experimental parameters in the laboratory are the same as in the field. It includes all main physical (e.g. radiation, humidity, air pressure), chemical (e.g. soil composition) and biological (e.g. growth rates) variables, not only temperature with freeze-thaw cycling. Please see the next questions.
Lines 118-126: How you verified that the results from the temperature-controlled chamber are statistically comparable to the field results?
Line 119: How were humidity and air pressure controlled in the chamber?
Line 123: Did you verify that your full-spetrum bulbs provide the same intensities and wavelengths as the Sun radiation at 5100 m.a.s.l.?
Lines 142-143: Is it realistic to simulate such concentrations of nutrients by Aeolian transfer in such an inhospitable environment?
Line 148: Why the nutrients supply was not distributed evenly or unevenly in smaller concentrations (like water) and was received only once?
Lines 150-151: Why the Controls were not sampled after the 2nd snowfall event?
Line 175: I agree that snow is generally considered as a distilled water. However, in a closer look, it is a multiphase system composed of air, ice crystals, trapped aerosol particles, and dissolved and absorbed gases. Moreover, it is also a porous medium through which air can diffuse. So strictly speaking, it is not exactly the same as deionized liquid water unless it melts. Have you ever compared the results with using sterile deionized water and using local snow, whether they are the same?
Line 224: The first appearance of the abbreviation OTU is not defined.
Lines 261-262: Why do you assume that the less than a week time frame of the freeze-thaw cycles is more realistic to field scenarios than 2 months cycling?
Line 280, Figure 3: Please discuss in more detail what is the reason for the statistically significant difference in a number of CFUs and relative abundance in lab microcosm compared to Field samples?
Lines 469-475: Why do you think cyanobacteria did not respond to water supplementation in contrast to Neochlorosarcina negevensis findings?
Lines 532-535: Please discuss in more detail what do you think are the reasons that only a small fraction, especially of the prokaryotic microbial community, is active under favorable environmental conditions?
Author Response
***Author responses are in bold and are preceded by ***
*** Thank you for reviewing our manuscript and for the helpful comments and questions which have improved our manuscript. We have responded to your feedback below.
My comments and questions:
Title and Abstract: The Title and the Abstract should contain information that most of the results come from laboratory experiments, not from the real (field) environment.
*** We partially agree with the reviewer. We made a couple changes in the Abstract to put more emphasis on the results obtained from lab microcosms (please see lines 17-18 and 28). However, since our work does contain field data, we believe the title is still accurate and does not need to be changed.
Abstract, Lines 27-29: “These results indicate that only a fraction of the detected community is able to become active under realistic favorable conditions, and that water alone can dramatically change community structure.” To make such a (realistic) conclusion, it is necessary to show that your experimental parameters in the laboratory are the same as in the field. It includes all main physical (e.g. radiation, humidity, air pressure), chemical (e.g. soil composition) and biological (e.g. growth rates) variables, not only temperature with freeze-thaw cycling. Please see the next questions.
*** We agree with the reviewer and we changed the sentence to: “These results indicate that only a fraction of the detected community is able to become active when water and nutrients limitations are alleviated in lab microcosms, and that water alone can dramatically change community structure” (please see line 28).
Lines 118-126: How you verified that the results from the temperature-controlled chamber are statistically comparable to the field results?
***It is not possible to statistically compare lab and field conditions since we did not attempt to fully mimic all of the field conditions. As discussed in the previous response, we wished to see which organisms responded if we alleviated nutrient and water limitations. The fact that field and lab experiments responded differently shows that some other factor in the field may also be limiting the ability of the community to respond or that the field experiment was not run for a long enough period of time (due to field logistical constraints). These limitations are discussed in the Discussion Section (please see lines 446-452).
Line 119: How were humidity and air pressure controlled in the chamber?
*** We did not control for humidity and air pressure in the chamber. The objective of the study was to understand which members of the endemic microbial communities would respond to realistic water and water + nutrient pulses mimicking the temperature fluctuations alone. We were not trying to simulate the full spectrum of environmental parameters.
Line 123: Did you verify that your full-spectrum bulbs provide the same intensities and wavelengths as the Sun radiation at 5100 m.a.s.l.?
*** We did not verify that our full-spectrum bulbs provide the same radiation output as that experienced by soil surfaces at 5100 m.a.s.l. on Volcán Llullaillaco. We did not try to mimic the exact intensities and wavelengths as those experienced in the field as our study focused on the response to water and water + nutrient pulses of microbial communities that are not directly exposed to surface radiation (4 cm depth).
Lines 142-143: Is it realistic to simulate such concentrations of nutrients by Aeolian transfer in such an inhospitable environment?
*** Our main objective was to overcome nutrient limitations and not to mimic a specific Aeolian event. We did not measure nutrient concentrations in the field following a specific Aeolian event for comparison. As we stated in Materials and Methods section, the concentrations used were based on preliminary laboratory experiments that determined that these amounts ensure an adequate amount to overcome nutrient limitations without changing the ionic balance of the soil solution (please see lines 150-152).
Line 148: Why the nutrients supply was not distributed evenly or unevenly in smaller concentrations (like water) and was received only once?
*** The reviewer brings up a good point about trying to distribute nutrient supplies in smaller amounts. We decided to provide a single nutrient input as we assumed that it would more likely for nutrients to be available following stochastic aeolian depositions from lower elevations rather than being distributed on a more frequent basis.
Lines 150-151: Why the Controls were not sampled after the 2nd snowfall event?
*** We did not have Controls for the 2nd snowfall event because of the amount of soil that was available to us to perform the experiment. In response to the reviewer we added to the text (please see lines 158-159)
Line 175: I agree that snow is generally considered as a distilled water. However, in a closer look, it is a multiphase system composed of air, ice crystals, trapped aerosol particles, and dissolved and absorbed gases. Moreover, it is also a porous medium through which air can diffuse. So strictly speaking, it is not exactly the same as deionized liquid water unless it melts. Have you ever compared the results with using sterile deionized water and using local snow, whether they are the same?
*** The reviewer certainly brings up a good point about the difference between using deionized sterile water and snow and future studies should take into account the amount of water that becomes available after a snow melt in the field, but there is currently no information about this at this site or any similar sites. Our study focused on the response of the indigenous community to simulated water inputs and we did not try to draw conclusions on how much melt occurs when snow is on the ground.
Line 224: The first appearance of the abbreviation OTU is not defined.
*** We agree with the reviewer and we have defined the first appearance of the abbreviation OTU (please see lines 236-237).
Lines 261-262: Why do you assume that the less than a week time frame of the freeze-thaw cycles is more realistic to field scenarios than 2 months cycling?
*** We assume that liquid water is available after a snowmelt only for a limited amount of time, on the order of days, rather than months, due to high sublimation rates in the environment (Schubert 2014). We wished to determine whether N. friedmannii would promptly take advantage of water availability on a short time scale. In response to the reviewer we have changed the text (please see lines 273 and 278-279).
Line 280, Figure 3: Please discuss in more detail what is the reason for the statistically significant difference in a number of CFUs and relative abundance in lab microcosm compared to Field samples?
*** The difference in the number of N. friedmannii CFUs between the field and microcosms can be explained by the multiple factors influencing water evaporation rate in the field, therefore water availability might have differed between field and microcosms. The more controlled environment in the laboratory microcosms may have allowed for more N. friedmannii cells to grow in response to the first water input. In response to the reviewer we have added to the text (please see lines 286-288).
Lines 469-475: Why do you think cyanobacteria did not respond to water supplementation in contrast to Neochlorosarcina negevensis findings?
*** We think that algae may outcompete cyanobacteria as primary producers in these environments because research has shown that cyanobacteria are relatively rare in melting snowfields and other non-stable cryosystems (Vincent 2000; Quesada and Vincent 2012).
In response to the reviewer we have added to the text (please see lines 494-495).
Lines 532-535: Please discuss in more detail what do you think are the reasons that only a small fraction, especially of the prokaryotic microbial community, is active under favorable environmental conditions?
*** We have added to the text to address the point brought up by the reviewer (please see lines 72-75).
Reviewer 2 Report
The manuscript “Limited responses of indigenous microbes to water and nutrient pulses in high-elevation Atacama soils: Implications for the cold-dry limits of life on Earth” by Drs. Vimercati and colleagues, reports very interesting findings on microbial life activation upon sudden hydration and nutrient supplementation, on a extremely cold dry environment. Most experiments seem to have been carefully conducted, and only few questions require clarification.
Abstract:
For the non-specialist reader, it would be interesting to have, at least once in the text, the “meters above sea level” initials described in full.
Introduction:
OTUs from some of the most extremely desiccation resistant bacteria commonly found in deserts namely Deinococcus, Rubrobacter, Geodermatophilus, etc, have not been mentioned in the Introduction. Was this because they were not detected in this study. Could this be a consequence of primer bias or merely absence from this ecosystem? A couple lines in the Discussion would help strengthen the uniqueness of this environment in this regard as well, as described for Cyanobacteria in section 4.2.2.
Materials and Methods:
It would be important to add a brief description of the culture media used after hydration and nutrient supplementation to recover bacterial and fungal CFUs. The media selection is of paramount importance for recovery of certain taxa, so, if only one or two were used, the diversity of recovered strains will also be low. For example, were media supplemented with trace vitamins?
It is incorrect to designate bacteria that we can’t culture in the lab as “not culturable” (line 2074)! This is simply a consequence of microbiologists’ ignorance about the complex conditions required for their growth, biotic, abiotic, etc. For example, was the incubator %O2 adjusted to mimic that at 5000 m.a.s.l., which may be close to 50% of the “normal” 21%? How about UV-light exposure, expectedly higher at 5100 m.a.s.l. (thinner atmosphere)?
Line 215. The plates in Supplementary Figure 3 show far less than 30-300 colonies. In fact, it is awkward that the plate selected to show the apparently dominant Naganishia isolates in that picture has only one single colony (as far as can be seen). Can this picture be replaced by one with more colonies? I am also curious about the methodology to confirm the true identity of Naganishia and about why the authors go to the species level based on OTU data alone (partial 18s gene)! How is microscopy considered suitable to identify this particular species and not another within the same genus? The authors cannot unequivocally identity Naganishia friedmannii (and Neochlorosarcina negevensis) from OTUs and morphology. To unequivocally confirm the identity of the reported results, maybe sequence the full 18s gene from a few colonies or additional genes to discriminate between species?
Maybe adding the meaning of a, b, ab, abc, etc, to the legend of Figure 4 and Supplementary Figure 3 would improve readability and facilitate interpretation.
When using primers for 16s regions as was the case in this study, and which can also “capture” some archaeal OTUs, I wonder if there were at least a few archaea detected? Is this a common trend for these environments or have some (rare) archaea already been detected in similar environments?
Line 457. Missing “of” between studies life.
Author Response
Author responses are in bold and are preceded by
Comments and Suggestions for Authors
The manuscript “Limited responses of indigenous microbes to water and nutrient pulses in high-elevation Atacama soils: Implications for the cold-dry limits of life on Earth” by Drs. Vimercati and colleagues, reports very interesting findings on microbial life activation upon sudden hydration and nutrient supplementation, on a extremely cold dry environment. Most experiments seem to have been carefully conducted, and only few questions require clarification.
*** Thank you for your review and for helping us clarify several points in our manuscript. Please see our responses below.
Abstract:
For the non-specialist reader, it would be interesting to have, at least once in the text, the “meters above sea level” initials described in full.
*** We agree with the reviewer and we have added “meters above sea level” to the initials m.a.s.l. the first time they appear in the main text (please see line 47).
Introduction:
OTUs from some of the most extremely desiccation resistant bacteria commonly found in deserts namely Deinococcus, Rubrobacter, Geodermatophilus, etc, have not been mentioned in the Introduction. Was this because they were not detected in this study. Could this be a consequence of primer bias or merely absence from this ecosystem? A couple lines in the Discussion would help strengthen the uniqueness of this environment in this regard as well, as described for Cyanobacteria in section 4.2.2.
*** The reviewer brings up an interesting point. 16S rRNA genes of some genera commonly found in deserts, such as Deinococcus, Rubrobacter, and Geodermatophilus, were retrieved in very low abundance in our study. Primers used in this study are commonly used to describe soil microbial communities, including deserts. We therefore think that the most likely explanation is that those genera were not found in high abundance in this study because of differences in environmental conditions in high-elevation deserts versus low-elevation deserts.
Materials and Methods:
It would be important to add a brief description of the culture media used after hydration and nutrient supplementation to recover bacterial and fungal CFUs. The media selection is of paramount importance for recovery of certain taxa, so, if only one or two were used, the diversity of recovered strains will also be low. For example, were media supplemented with trace vitamins?
*** The goal of the culturing experiment was to specifically confirm that Naganishia cells were viable in the soils used in the experiments and to supplement the relative abundance data, which we clarified in lines 211-212. We therefore used media known to be suitable to grow Naganishia (Vimercati et al. 2016), which we clarified in lines 217-218. We did not try to culture other genera but Actinomycete bacteria and Moesziomyces fungi also happened to grow on this medium (lines 228-229). The medium was not supplemented with trace vitamins and it is described in Vimercati et al. 2016. In response to the reviewer we have changed the text (please see lines 211-212).
It is incorrect to designate bacteria that we can’t culture in the lab as “not culturable” (line 2074)! This is simply a consequence of microbiologists’ ignorance about the complex conditions required for their growth, biotic, abiotic, etc. For example, was the incubator %O2 adjusted to mimic that at 5000 m.a.s.l., which may be close to 50% of the “normal” 21%? How about UV-light exposure, expectedly higher at 5100 m.a.s.l. (thinner atmosphere)?
***We agree with the reviewer that most Bacteria don’t grow in the lab because of lack of knowledge on what they require to grow and thus have deleted that statement. We did not adjust the %O2 or UV exposure in the chamber to mimic the exact conditions in the field as we were not trying to simulate the full spectrum of environmental parameters. Our objective was to grow N. friedmannii cells (see answer to above question). In response to the reviewer we have deleted a sentence (please see line 212) and changed the text (please see lines 228-229).
Line 215. The plates in Supplementary Figure 3 show far less than 30-300 colonies. In fact, it is awkward that the plate selected to show the apparently dominant Naganishia isolates in that picture has only one single colony (as far as can be seen). Can this picture be replaced by one with more colonies? I am also curious about the methodology to confirm the true identity of Naganishia and about why the authors go to the species level based on OTU data alone (partial 18s gene)! How is microscopy considered suitable to identify this particular species and not another within the same genus? The authors cannot unequivocally identity Naganishia friedmannii (and Neochlorosarcina negevensis) from OTUs and morphology. To unequivocally confirm the identity of the reported results, maybe sequence the full 18s gene from a few colonies or additional genes to discriminate between species?
*** We appreciate this comment but unfortunately, we do not have any other pictures than those shown in the supplementary material. The goal of the picture was to show an isolated Naganishia colony and not to show abundance. The picture shows the plate right after the first colony was visible, so other colonies have not appeared yet. We have added this point to the end of the caption.
We agree with the reviewer that sequencing of the full 18S rRNA gene would be needed to unequivocally confirm the identity of N. friedmannii. We believe it is extremely unlikely for those colonies to be another microeukaryote because N. friedmannii is the most relative abundant (> 90%) eukaryotic organism that 18S rRNA gene surveys have identified in the soils used in this study (Lynch et al. 2012; Vimercati et al. 2016).
Maybe adding the meaning of a, b, ab, abc, etc, to the legend of Figure 4 and Supplementary Figure 3 would improve readability and facilitate interpretation.
*** We stated in the captions of both figures that that different letters signify significant pairwise comparisons (Tukey HSD, p < 0.05) (please see lines 328-328 and 3rd-4th lines in Supplementary Figure 3).
When using primers for 16s regions as was the case in this study, and which can also “capture” some archaeal OTUs, I wonder if there were at least a few archaea detected? Is this a common trend for these environments or have some (rare) archaea already been detected in similar environments?
*** Thank you for this interesting question. Our 16S rRNA gene survey detected a few archaeal OTUs belonging to the Nitrososphaeraceae family (Thaumarchaeota). Members of Thaumarchaeota have been recovered at high elevations on this Volcano before (Lynch et al. 2012). We decided not to discuss the presence of this taxon in this manuscript because of their very low abundance.
Line 457. Missing “of” between studies life.
*** We added the word “of” as suggested by the reviewer (please see line 477).
Reviewer 3 Report
The manuscript entitle -Limited response of indigenous microbes to water and nutrient pulses in high-elevation Atacama soils: implications for the cold-dry limits of life on Earth- by Vimercati et al. describes how the addition of water and nutrients to high-elevation soils from Volcán Llullaillaco, both in the field and in the reconstructed lab microcosms, could affect the microbial communities. The manuscript is well written and easy to ready. The introduction provides sufficient background and includes relevant and recent references; the research design is appropriate and the methods are adequately described. Moreover, the results are clearly showed, the experimental data are exhaustive and the conclusions of authors are in lines with the results reported.
Since high-elevation desert soils are also of great interest to the field of Astrobiology, taking in account that the actual climate of Mars can be compared with that of a very high mountain on Earth such as the high Andes, this manuscript can be accepted for publication in this special issue after minor corrections.
Please shift the fig 2 from M&M section to Results section.
Supplementary fig. 3, fig. 4. please leave just the legend sof both figures and insert any comments in the main text of manuscript.
Author Response
***Author responses are in bold and are preceded by ***
Comments and Suggestions for Authors
The manuscript entitle -Limited response of indigenous microbes to water and nutrient pulses in high-elevation Atacama soils: implications for the cold-dry limits of life on Earth- by Vimercati et al. describes how the addition of water and nutrients to high-elevation soils from Volcán Llullaillaco, both in the field and in the reconstructed lab microcosms, could affect the microbial communities. The manuscript is well written and easy to ready. The introduction provides sufficient background and includes relevant and recent references; the research design is appropriate and the methods are adequately described. Moreover, the results are clearly showed, the experimental data are exhaustive and the conclusions of authors are in lines with the results reported.
Since high-elevation desert soils are also of great interest to the field of Astrobiology, taking in account that the actual climate of Mars can be compared with that of a very high mountain on Earth such as the high Andes, this manuscript can be accepted for publication in this special issue after minor corrections.
*** Thank you for your positive review of our manuscript.
Please shift the fig 2 from M&M section to Results section.
*** Figure 2 from the Materials and Methods section will be moved to the Results section in the published version of the Manuscript.
Supplementary fig. 3, fig. 4. please leave just the legend sof both figures and insert any comments in the main text of manuscript.
*** The caption of Supplementary Figure 4 is quite short and directly relevant for
interpreting the figure and thus we think it should remain below the figure. We agree that
the Supplementary Figure 3 caption is long. However, this information would make the
paragraph describing the culturing methods quite long and we prefer to keep it in the Supplementary Material.